# Dating App Use and Wellbeing: An Application-Based Pilot Study Employing Ecological Momentary Assessment and Objective Measures of Use

**DOI:** 10.3390/ijerph20095631

**Published:** 2023-04-25

**Authors:** Gabriel Bonilla-Zorita, Mark D. Griffiths, Daria J. Kuss

**Affiliations:** 1International Gaming Research Unit, Psychology Department, Nottingham Trent University, 50 Shakespeare Street, Nottingham NG1 4FQ, UK; 2Cyberpsychology Research Group, Psychology Department, Nottingham Trent University, 50 Shakespeare Street, Nottingham NG1 4FQ, UK

**Keywords:** dating applications, online dating, wellbeing, ecological momentary assessment, multilevel analysis

## Abstract

Smartphones are part of individuals’ daily lifestyles, as are smartphone applications such as dating apps. Previous evidence suggests that high engagement in dating applications can be detrimental to some users’ wellbeing. However, much of the published research has relied on cross-sectional studies and self-report measures. Therefore, the present study aims to overcome the limitations of subjective measures in cross-sectional designs by investigating for the first time the relationship between dating app users’ wellbeing (self-esteem, craving and mood) and objective measures of their use during a one-week period. To do this, the present study employed a newly developed application, *DiaryMood* and utilized ecological momentary assessment (EMA), as it measured subjects’ mood, self-esteem and craving three times per day and the time spent using the dating apps per day during one week. A convenience sample of 22 online dating app users participated in the present study. Findings from a three-level multilevel analysis indicated that increased time spent on dating apps predicted craving among dating app users and that notifications led to an improved mood and higher self-esteem. The results are discussed in relation to previous online dating studies. In sum, the present study sets a precedent for the use of EMA within the scope of online dating research, which may promote further studies adopting this methodology.

## 1. Introduction

Approximately 84% of the world’s population (6.64 billion individuals) owns a smartphone [1]. Consequently, many computer-based services (e.g., gaming, social media, online dating) have become ubiquitous due to the appearance of smartphone-based applications. However, having constant access can lead to potentially negative consequences for a minority of individuals. For instance, higher availability has been related to problematic use of social media networks [2,3] and dating applications [4]. Furthermore, it was reported that online dating users’ behavior changed when shifting from computer-based online dating to smartphone-based dating, resulting in higher engagement with dating applications [5]. Problematic use of online dating has been previously characterized [6] based on the components model of addiction [7], which comprises six components (i.e., salience, mood modification, tolerance, conflict, withdrawal, and relapse). Although problematic use of online dating does not currently constitute a mental health disorder diagnosis in any of the diagnostic manuals, there is growing empirical evidence that relates problematic use of online dating to lower psychological wellbeing and depression [8,9], as well as lower levels of self-esteem and body satisfaction [10].

Relatedly, loneliness and/or relatedness needs have been raised in previous studies as predictors of higher dating app engagement and problematic use [6,11]. In addition, previous findings have reported that needs-driven use is a significant predictor of higher dating app use [11,12]. More specifically, users reported that receiving matches and likes from other users was perceived as a form of (short-term) gratification (i.e., a self-esteem boost). Similarly, receiving smartphone notifications has been associated with the emotional states of the users [13]. For instance, receiving numerous notifications has been found to relate to negative emotional states (e.g., lower mood). However, if such notifications came from social networking sites, users felt socially connected and experienced a positive emotional state (i.e., better mood) [14]. 

Nevertheless, the number of notifications received can have an effect on users’ wellbeing, irrespective of the notification source (i.e., social or not), because high numbers of notifications can lead users to feel overloaded and experience decreased wellbeing [15], causing fatigue and self-esteem deterioration [16,17]. Notifications can also trigger the fear of missing out (FOMO) [18], which has been defined as the *“pervasive apprehension that others might be having rewarding experiences from which one is absent”* [19] (p. 1841). Previous research has reported that FOMO is a significant predictor for the maintenance of dating app usage behavior [20,21], in line with previous research that found FOMO to be a predictor of social media addiction and lower wellbeing [3]. Dating app users have identified feelings of FOMO when not active on the apps, and FOMO was also found to be influenced by structural characteristics of dating apps [20]. Furthermore, FOMO can lead smartphone users to experience increased feelings of craving [22] and repeatedly check their screens not to miss out on messages [18], which in turn can facilitate constant screen checking becoming a habit [23]. 

In line with this, the Interaction Person-Affect-Cognition-Execution (I-PACE) model [24] posits that individuals with a vulnerability to online addictions behave predominantly by impulse/reaction in response to internal/external stimuli (i.e., triggers), which inhibits self-regulatory control over urges. Consequently, screen-checking behavior could become conditioned as a coping mechanism to overcome negative emotional states. Moreover, the revised I-PACE model [24] differentiates between early and later stages of online addiction; in the early stages, the individual is guided primarily by the gratification and reward obtained through engagement in the activity (i.e., online dating). However, in later stages, craving and cue-reactivity are key to maintaining dating app use by primarily obtaining compensation and further reinforcement of generated affective and cognitive biases and coping mechanisms.

Most of the past research examining online addictions, and more specifically, problematic use of online dating, has relied upon self-report methodologies. For instance, in a review of the published studies in social psychology in the year 2018, it was reported that 68% of the published studies relied exclusively on self-report measures [25]. This could present a problem given that self-report data have been found to lack accuracy when participants report their own use of social media [26,27], which can lead to both overreporting and underreporting of findings [28]. However, ecological momentary assessment (EMA) is a sampling technique that collects real-time data in participants’ natural settings, decreasing recall bias and promoting ecological validity [29]. Contrary to self-report scales that aim to obtain an overall estimate of a given construct, EMA is able to register those changes in participants’ behavior and/or general wellbeing throughout the study period [30]. Furthermore, given the widespread use of smartphones, carrying out EMA studies is simpler than in the past, when participants needed to carry additional items to log their behavior (e.g., paper and pencil) [30]. It is now possible for participants to log onto their smartphones and register their responses in real time. Moreover, the use of smartphones to carry out such studies allows the possibility of “passive monitoring”, which means that the data are collected automatically (e.g., screen time, number of screen unlocks) without the need for participants’ recall [31,32].

Previous findings regarding media addictions and dating app research have highlighted relationships between the number of notifications and users’ wellbeing (i.e., mood and self-esteem), frequent checking of smartphones with the development of habitual usage and increased feelings of craving, and high-frequency dating app use with lower mental health and general wellbeing. The present study investigated the relationship between wellbeing measures, including self-esteem, mood, and craving, and objective measures of dating app use (i.e., usage time, number of notifications, number of launches). To do this, a newly developed smartphone application was employed to collect real-time data from participants (i.e., wellbeing measures and objective measures of use). It was hypothesized that higher usage time on dating apps would lead to lower mood (*H*_1_) and lower self-esteem (*H*_2_), as well as a higher craving to be on an online dating app (*H*_3_). It was also hypothesized that notifications would lead to higher cravings to be on an online dating app (*H*_4_), increased mood (*H*_5_), and increased self-esteem (*H*_6_). Finally, it was hypothesized that the number of launches (i.e., screen unlocks) would lead to decreased mood (*H*_7_), decreased self-esteem (*H*_8_), and a higher craving to be on an online dating app (*H*_9_).

## 2. Methods

### 2.1. Design

The study consisted of real-time self-reported repeated measures collected using a newly developed smartphone app (i.e., *DiaryMood*) in which participants responded to questions regarding the following areas three times a day: (i) mood, (ii) self-esteem, and (iii) craving (i.e., in the morning, afternoon, and evening). In addition, participants included their daily use of dating applications, the number of launches (i.e., the number of times participants opened the application), and the number of notifications they received from dating applications. Participants were advised to set alarms on their smartphones to complete questions during each measurement timepoint. Additionally, calendar reminders were also scheduled via the email addresses that participants used to express their interest in taking part in the study to ensure completion of the measures. In order to participate, participants were required to be at least 18 years old and current users of at least one online dating application. The study required participants to record each of the measures for seven consecutive days (i.e., one full week), and it required a few seconds to respond to each measure across the three timepoints (~20 s). Although participants needed to be contacted via email to participate in the study, the data from participants were anonymized so that their emails were not associated with their data. To do this, once participants stated their interest in participating, they were given a unique code and password for their access to the app. Once they launched the application (i.e., signed-in), they were asked to create a unique code that only they knew in case a participant wanted to remove their data from the study and to keep complete anonymity (as stated in the ethical approval for the study). In order to increase participation, the study offered a compensation of a £20 Amazon voucher, approved by the research team’s university ethics committee. Participants received an information form and a consent form after stating their interest in participating. Once they had signed the consent form, they were sent the link to download *DiaryMood* onto their smartphones. Once the study finished, participants were given a debriefing form and the link to their voucher.

### 2.2. Participants

A total of 22 participants took part in the study (*M*_age_ = 24.82 years, *SD* = 4.36). Participants were recruited through social media networks (e.g., Facebook, Instagram), where the study was posted. Further participants were recruited through the university’s research credit participation system. Participation was voluntary, and participants contacted the first author to express their interest in taking part in the study. In order to be eligible for the study, participants needed to (i) be at least 18 years old, (ii) be current users of dating apps, and (iii) be Android users. Further details on the participants’ socio-demographics can be found in Table 1.

### 2.3. Materials

To collect the data, an Android-based application (*DiaryMood*) was developed to include the measures for the present study. *DiaryMood* included sociodemographic items (i.e., age, gender, sexual orientation, nationality, and occupation). Regarding the measures, *DiaryMood* included three items concerning mood, self-esteem, and craving. Each of the items was presented on a single screen, where participants needed to tap on one of the options and press ‘continue’ afterwards. For mood, participants responded to the following item: “Rate your mood” on a Likert scale ranging from 1 (extremely unhappy) to 5 (extremely happy) [33]. For self-esteem, the item read “Rate your self-esteem: I have high self-esteem” from 1 (not very true of me) to 5 (very true of me) [34]. For craving, the item read “How much would you like to be on your dating app right now?” on a scale from 1 (not at all) to 5 (very much) [35]. For the objective measures, participants logged their responses on a tab that read “*Log your stats of use*”. When clicking on the tab, participants were presented with three boxes that included their daily use of dating applications, total use time (in minutes), number of notifications, and number of launches. To access the objective measures, participants were asked to collect data from the wellbeing section on their Android smartphones. For a visual example of *DiaryMood,* see Figure 1 and Figure 2.

### 2.4. Statistical Analysis

Analysis was carried out in RStudio (version 1.2.1335, Boston, MA, USA). First, descriptive statistics were analyzed with regard to sample demographics, means, and standard deviations of the study variables (Table 2). Subsequently, Pearson’s correlations were calculated to assess the correlations between the variables of the study (Table 3). Considering that the study aimed to analyze the variation within each individual over time regarding their wellbeing and dating app use and the variation between individuals, multilevel analysis was performed to assess these relationships between wellbeing variables (i.e., outcome variables) and objective measures (i.e., predictive variables). To do this, the three daily measures (Level 1) were nested within days (Level 2) within participants (Level 3). An example of this three-level model is shown in Figure 3. The data were ordered so every participant’s data started on a Monday and ended on a Sunday to control for possible patterns of usage/wellbeing based on the day of the week (see Figure 4, Figure 5, Figure 6 and Figure 7). Further analyses were carried out to obtain standardized estimates and 95% confidence intervals with ‘effectsize’ package [36]. As expected in an EMA study [37], there were missing datapoints that appeared to be missing at random (MAR). Therefore, treatment of missing data was handled by the default option of the ‘lmer’ function from the ‘lm4’ package [38], which excludes rows containing missing datapoints, as according to Snijders and Bosker (1999) [39], this does not lead to biased estimates if the condition of MAR is met. 

## 3. Results

Results suggested that mood and self-esteem levels across the study week remained stable within a medium-high range (*M*_mood_ = 3.39, *SD*_mood_ = 0.95; *M*_self-esteem_ = 3.39, *SD*_self-esteem_ = 1.12) with a small divergence during the weekend when mood was slightly higher than self-esteem (see Figure 4 and Table 2 for descriptive statistics). In the case of craving, participants were within the medium range (i.e., 2–2.5; see Figure 4), with Wednesday the only day that craving levels surpassed the medium point (*M*_craving-Wednesday_ = 2.59). Usage was highest at the start of the week, while differences were not statistically significant. Tuesday’s average use was 41.68 min (the highest during the week). The second highest day of use was Thursday with an average of 35.59 min, followed by Saturday with 33.18 min (see Figure 5). Regarding number of notifications, Tuesday was the day with the highest number of dating app notifications received, with an average of 58.62, followed by Saturday with an average of 48.36 notifications (see Figure 7). In the case of number of launches, Saturday was the day with the highest average number of launches of dating applications with 32.27, and the second highest day was Tuesday with 25.58 launches (see Figure 6). The intraclass correlation coefficients (*ICCs*) suggested that 55% of the variance in launch averages was explained by between-participant variation. Therefore, 45% corresponds to within-participant variation, indicating that the difference was higher between participants’ numbers of launches than the differences in launches within participants. In the case of craving, 18% of the variance was attributed to between-participant variance and 82% to within-participant variance, indicating that each participant’s level of craving differed across the week more than the difference found between each other’s levels of craving (see Table 2).

Associations between variables are shown in Table 3. Mood and self-esteem were more strongly correlated (*r* = 0.77, *p* < 0.001) than self-esteem and usage (*r* = 0.12, *p* < 0.05), and mood and launches (*r* = 0.12, *p* < 0.05). In addition, objective measures (i.e., usage, launches, and notifications) showed strong correlations with each other: notifications and launches (*r* = 0.66, *p* < 0.001), notifications and usage (*r* = 0.75, *p* < 0.001), and usage and launches (*r* = 0.72, *p* < 0.001).

Three models, one for each outcome variable (i.e., mood, self-esteem, and craving) were tested. Each of the models was compared against alternative models in terms of their fit indexes (i.e., AIC, BIC, and deviance). The resulting models and their fit indexes are presented in Table 4. The model fit for mood as the outcome variable (i.e., Model 1) with random intercepts and random slopes was found to have the best fit (AIC = 851.3, BIC = 892.7, and deviance = 829.3). For self-esteem as the outcome variable (Model 2), random intercepts and random slopes were found to have the best fit (AIC = 921.2, BIC = 962.5, and deviance = 899.2). For the best fit of the model with craving as the outcome variable, random intercepts and random slopes were found to have the best fit (AIC = 935.6, BIC = 976.9, and deviance = 913.6). Reaching the level of statistical significance, it was found that for every one-unit increase in notifications, participants’ mood increased by 0.14 (β = 0.14, *p* = 0.014). In the case of self-esteem, for every one-unit increase in notifications, self-esteem increased by 0.23 (β = 0.23, *p* = 0.006). For craving, it was found that for every on-unit increase in usage, craving increased by 0.19 (β = 0.19, *p* = 0.044). Further results from the three models are presented in Table 5, Table 6 and Table 7.

## 4. Discussion

The present study investigated the relationships between objective measures of dating app use (i.e., use time, number of launches, and number of notifications) and users’ wellbeing (i.e., mood, self-esteem, and craving) during a one-week period. To do this, a smartphone-based application for Android phones (i.e., *DiaryMood*) was developed. The study collected the data in participants’ natural settings and registered 12 daily responses per participant in real time, based on the principles of EMA [29]. 

According to the MLM results, no significant effect was found for the time spent on dating applications (i.e., use time) on mood or self-esteem. Therefore, neither *H*_1_ nor *H*_2_ were supported. Contrary to this, other studies have found lower scores on wellbeing measures (i.e., depression and anxiety) in relation to higher use of online dating apps [9,40,41] and lower self-esteem when comparing users and non-users of the dating application Tinder [10]. Nevertheless, these studies measured online dating use by frequency of log-ins and/or retrospective self-report measures, which may lead to different results in comparison to actual time spent using the app, as used in the present study. Conversely, other scholars have found positive outcomes in terms of users’ wellbeing and dating app use. For example, Watson et al. (2019) [42] reported that dating app users felt emotional connectedness as a result of their use, which is in line with findings that claim that users experienced increased wellbeing when they received matches or met new individuals on dating apps [43]. 

In relation to craving and use of time, a significant association was found. Therefore, *H*_3_ was supported. More specifically, higher dating app use time predicted higher levels of craving. Related to this finding, Hormes et al. (2014) [44] reported that users addicted to social media (according to modified alcohol dependence criteria from the DSM-IV-TR [45], in which craving is included as a criterion) used Facebook substantially more than initially intended and yet experienced high levels of craving for Facebook. Additionally, craving to use dating apps may be another step to provide evidence regarding the problematic use of dating apps, given that craving has been identified as a key construct in the pathophysiology of behavioral addictions in the DSM-5 [46]. Furthermore, cue-induced craving has been found to predict internet-communication disorder [47]. Considering that smartphones can be a craving-inducing cue [48] for dating applications and their constant presence in the daily lives of users, it is likely that the association cycle between habitual behaviors, and cognitive and emotional responses will become stronger [24]. 

In the case of notifications and craving, no significant relationship was found in the MLM analysis. Therefore, *H*_4_ was not supported by the MLM. Previous literature has suggested that notifications can act as reminders of activity and increase feelings of FOMO [18]. Receiving notifications of messages, matches, or likes can act as cues inducing craving for dating app use [47]. Moreover, some studies have found that social-based notifications lead to positive emotional states [13,14], which is in line with *H*_5_ and *H_6,_* which indicate that notifications would be associated with better mood and self-esteem, as supported by the findings in the MLM analysis. According to these findings, dating app users experienced a positive outcome when they received dating app notifications, which is in line with previous findings where participants reported using dating applications to fulfil their short-term needs [8,41,49]. Furthermore, in previous research, relatedness frustration was found to significantly predict higher online dating intensity. Considering these findings, experiencing better mood and self-esteem when receiving notifications may be explained by the expectation that users of dating apps will meet their needs. Arguably, if a given user’s goal is to receive social and/or romantic attention from other users, receiving message notifications can be considered the signal of accomplishment of such a goal, leading to positive emotional states. Another explanation may be that notifications could have been conditioned to positive outcomes such as need gratification (i.e., classical conditioning). Therefore, further studies should assess the interaction between types of notifications (e.g., matches vs. automatically-generated notifications) and users’ wellbeing.

Regarding the number of launches, there were no significant findings in the MLM analysis for mood (*H*_7_), self-esteem (*H*_8_), or craving (*H*_9_). Oulasvirta et al. (2012) [48] reported that habitual checking of the smartphone was not considered negatively by users. In fact, users reported positive outcomes from repetitive checking, such as time-killing and entertainment. For instance, the highest number of launches throughout the week happened on Saturday, which may have facilitated users meeting in person and potentially improving their wellbeing. In the case of craving, launching dating applications could lead to cue-reactivity and increased feelings of craving, as studies examining cybersex addiction have shown [50,51]. Nevertheless, the relationships between the number of launches and wellbeing measures were not supported in the present study. Therefore, future studies should further assess the frequency of checking dating applications and their relationship with subjective feelings of wellbeing. 

## 5. Limitations

The present study is not without limitations. First, the small sample size (*N* = 22) may have reduced the statistical power to find significant effects. Second, the sample was collected via convenience sampling; therefore, the findings cannot be generalized to the general population of online dating users [52]. Third, in order to facilitate data collection, participants were not given specific times for when to fill in their responses, although they were advised to respond in the morning when they wake up, in the afternoon (12:00–13:00), and in the evening (from 20:00 to their bedtime), and to set smartphone alarms with the advised times. Fourth, for ten participants, English was not their mother tongue, and although they were informed and assisted with the language barrier (if needed), some responses might have been biased or misrepresented. Fifth, the variables of mood, craving, and self-esteem have been assessed with one item, with the limitations that this entails. However, considering this is a pilot study, the authors believe that using single items for each of the wellbeing variables is justified to provide preliminary evidence. All in all, the present study provides novel evidence in the field of online dating, and it is innovative in the use of (i) a smartphone-based application to carry out data collection within the scope of online dating research, and (ii) EMA methodology to include objective measures of dating app use. 

## 6. Conclusions

The present study assessed the relationships between objective measures of dating app use (i.e., use time, notifications, and launches) and users’ wellbeing. Participants responded to daily questions for seven days utilizing the *DiaryMood* app, which was designed for the purpose of the present study. Overall, the present study provides new evidence in the study of problematic dating app use. More specifically, findings from the study highlight the relevance of dating app notifications in relation to users’ wellbeing. In addition, the finding that increased time spent on dating apps predicted craving for dating app use provides preliminary evidence that can be used for further study of potential addiction to dating applications. Moreover, the present study represents, to the best of the present authors’ knowledge, the first study to employ ecological momentary assessment within the field of problematic use of online dating and provides new evidence on the potentially addictive dynamics that may underlie problematic use of dating applications. It is hoped that findings of the present study (i) will promote further research employing objective methods, (ii) will provide evidence that apps such as *DiaryMood* are advantageous tools to carry out empirical studies on online addictions, and (iii) will provide further evidence in the study and conceptualization of problematic use of online dating.

## Figures and Tables

**Figure 1 ijerph-20-05631-f001:**
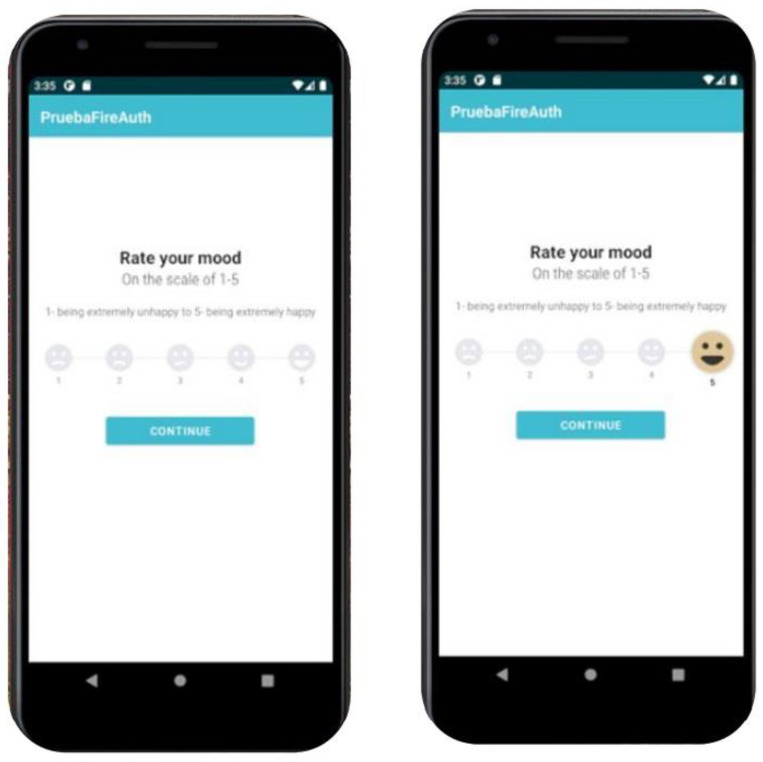
*MoodDiary* mood item.

**Figure 2 ijerph-20-05631-f002:**
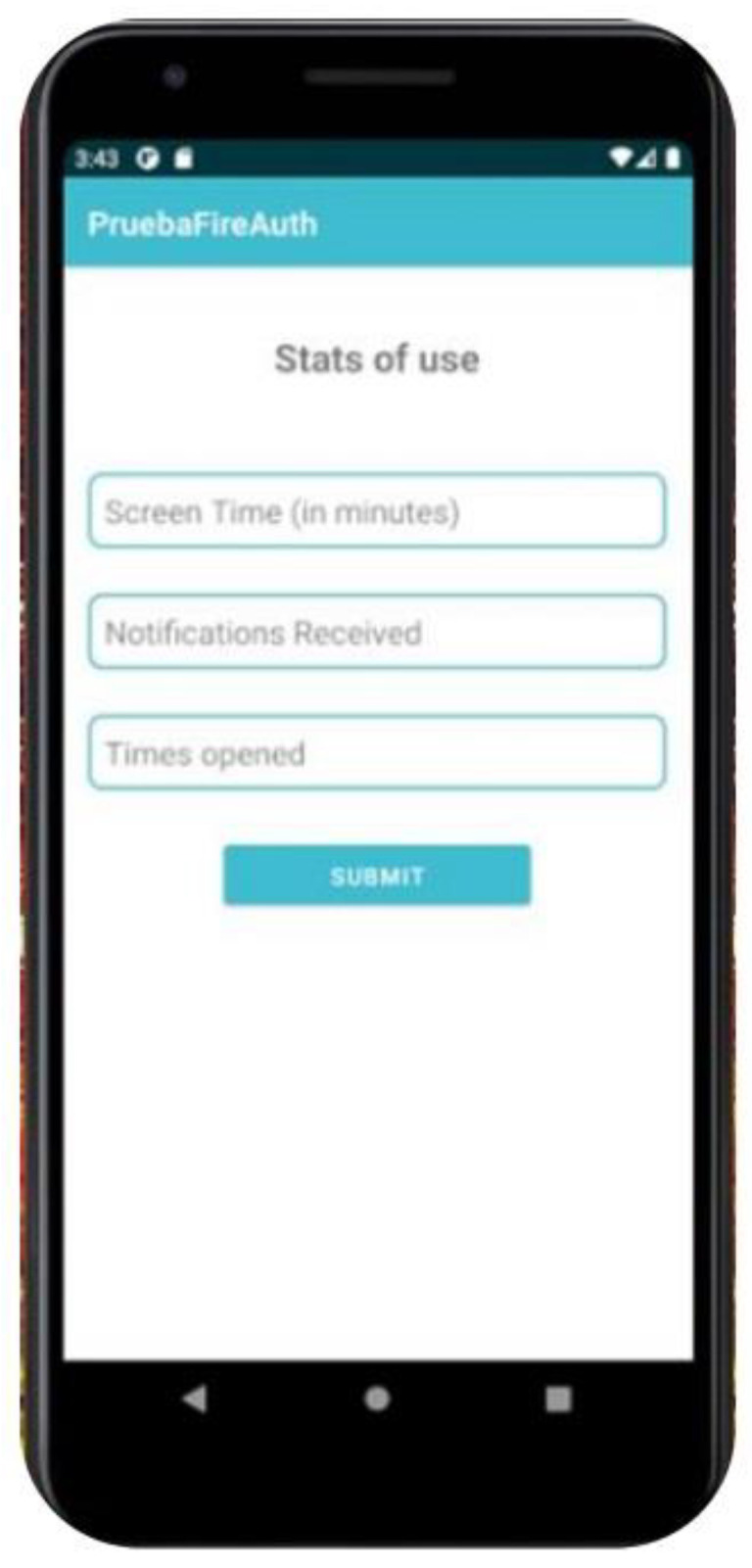
*MoodDiary* objective measures.

**Figure 3 ijerph-20-05631-f003:**
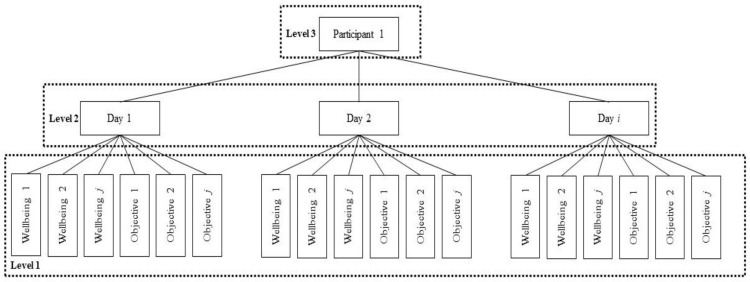
Hierarchical structure of the three-level model.

**Figure 4 ijerph-20-05631-f004:**
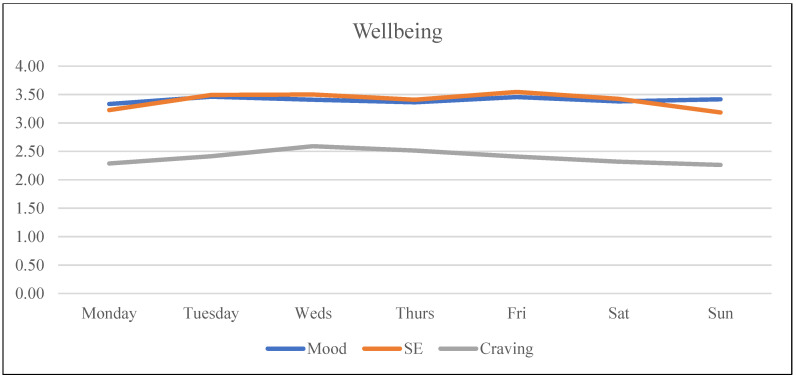
Mood, self-esteem, and craving across the week.

**Figure 5 ijerph-20-05631-f005:**
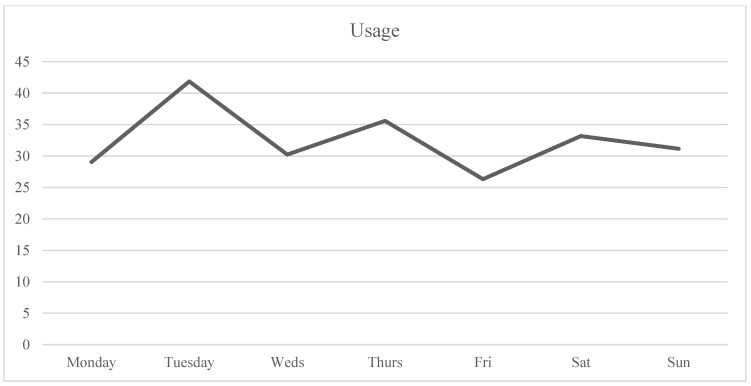
Use time across the week.

**Figure 6 ijerph-20-05631-f006:**
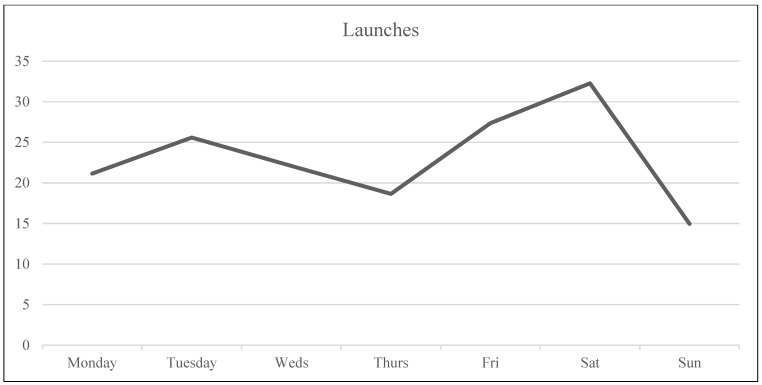
Number of launches across the week.

**Figure 7 ijerph-20-05631-f007:**
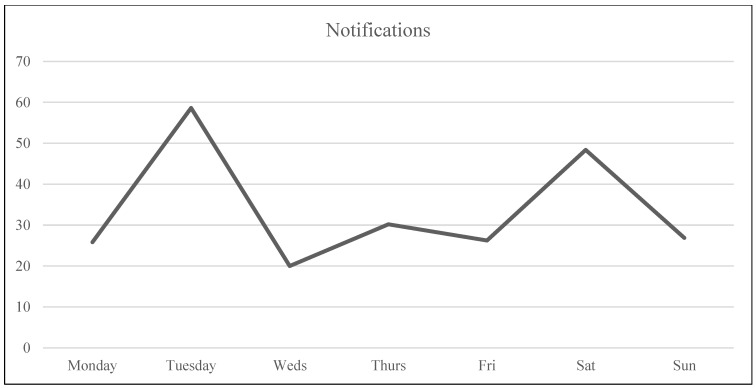
Number of dating apps’ notifications across the week.

**Table 1 ijerph-20-05631-t001:** Demographics of the total sample, *N* = 22.

	*n* (%)
Age (mean, *SD*)	24.82 (4.36)
Gender	
Female	16 (72.7)
Male	6 (27.3)
Sexual orientation	
Heterosexual	14 (63.6)
Homosexual	3 (13.6)
Bisexual	5 (22.7)
Marital status	
Single	21 (95.5)
In a relationship	1 (4.5)
Occupation	
Student	12 (54.5)
Full-time job	6 (27.3)
Part-time job	2 (9.1)
Freelance	2 (9.1)

**Table 2 ijerph-20-05631-t002:** Descriptive statistics.

	Mean	*SD*	*ICC*
Mood	3.39	0.95	0.36
Self-esteem	3.39	1.12	0.47
Craving	2.42	1.11	0.18
Usage (in minutes)	28.04	31.37	0.40
Notifications	25.42	67.35	0.34
Launches	18.79	25.36	0.55

*SD*, standard deviation; *ICC*, interclass correlation coefficient.

**Table 3 ijerph-20-05631-t003:** Correlation matrix of study variables.

	Mood	Self-Esteem	Craving	Usage	Notifications	Launches
Mood	-					
Self-esteem	0.77 ***	-				
Craving	0.07	0.14 **	-			
Usage	0.08	0.12 *	0.20 ***	-		
Notifications	0.15 **	0.16 **	0.14 **	0.75 ***	-	
Launches	0.12 *	0.16 **	0.15 **	0.72 ***	0.66 ***	-

* *p* < 0.05; ** *p* < 0.01; *** *p* < 0.001.

**Table 4 ijerph-20-05631-t004:** Model fit statistics.

	AIC	BIC	Deviance	LogLik
Model 1 (Mood)	851.3	892.7	829.3	−414.7
Model 2 (Self-esteem)	921.2	962.5	899.2	−449.6
Model 3 (Craving)	935.6	976.9	913.6	−456.8

AIC, Akaike information criterion; BIC, Bayesian information criterion; LogLik, Log-likelihood.

**Table 5 ijerph-20-05631-t005:** Mood as outcome (Model 1).

	B	SE	β	*p*-Value	Standardized 95% CI
Intercept	30.35	0.10	0.00	<0.001 ***	[0.00, 0.00]
Usage	−0.001	0.003	−0.05	0.548	[−0.23, 0.12]
Launches	0.0004	0.003	0.01	0.905	[−0.16, 0.18]
Notifications	0.003	0.001	0.21	0.014 *	[0.05, 0.38]
Random effects	Variance	*SD*			
Participants: Day (Intercept)	0.43	0.66			
Day (Intercept)	0.00	0.00			
Residual	0.54	0.73			

B, coefficient estimate; SE, standard error; β, standardized correlation coefficient; CI, confidence interval; *SD*, standard deviation. * *p* < 0.05; *** *p* < 0.001.

**Table 6 ijerph-20-05631-t006:** Self-esteem as outcome (Model 2).

	B	SE	β	*p*-Value	Standardized 95% CI
Intercept	30.37	0.11	0.00	<0.001 ***	[0.00, 0.00]
Usage	−0.003	0.003	−0.08	0.328	[−0.24, 0.08]
Launches	0.001	0.004	0.03	0.716	[−0.13, 0.19]
Notifications	0.004	0.001	0.23	0.006 **	[0.08, 0.39]
Random effects	Variance	*SD*			
Participants: Day (Intercept)	0.86	0.93			
Day (Intercept)	0.0002	0.01			
Residual	0.568	0.75			

B, coefficient estimate; SE, standard error; β, standardized correlation coefficient; CI, confidence interval; *SD*, standard deviation. ** *p* < 0.01; *** *p* < 0.001.

**Table 7 ijerph-20-05631-t007:** Craving as outcome (Model 3).

	B	SE	β	*p*-Value	Standardized 95% CI
Intercept	20.34	0.10	0.00	0.000 ***	[0.00, 0.00]
Usage	0.01	0.003	0.19	0.044 *	[0.01, 0.38]
Launches	−0.001	0.003	−0.03	0.730	[−0.19, 0.13]
Notifications	0.0001	0.001	0.01	0.894	[−0.18, 0.20]
Random effects	Variance	*SD*			
Participants: Day (Intercept)	0.21	0.45			
Day (Intercept)	0.00	0.01			
Residual	0.93	0.96			

B, coefficient estimate; SE, standard error; β, standardized correlation coefficient; CI, confidence interval; *SD*, standard deviation. * *p* < 0.05; *** *p* < 0.001.

## Data Availability

The data are not publicly available due to privacy and ethical reasons.

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
