# Peer review of "Dating App Use and Wellbeing: An Application-Based Pilot Study Employing Ecological Momentary Assessment and Objective Measures of Use"

_ijerph, 2023, doi:10.3390/ijerph20095631_

Round 1

Reviewer 1 Report (Previous Reviewer 2)

Thank you for the point by point response.I have no further comments.

This manuscript is a resubmission of an earlier submission. The following is a list of the peer review reports and author responses from that submission.

Round 1

Reviewer 1 Report

I read with great interest the results of the research project presented in the article. I find the relatively innovative methodological approach interesting. The article is mature, written in a language understandable to a wider range of readers. It is rather a research report (taking into account the size of the sample, it can be considered a pilot study), hence the scientific and cognitive value is modest. Nevertheless, I treat this article as a presentation of a good foundation for further exploration. If I had to suggest any additions to the authors, they would concern few aspects.

First, the purpose of the research and the article should be more clearly emphasized in the abstract. The current form of the abstract does not fully reflect the content of the article.

Secondly, the adopted methodology requires better justification. Although the authors indicated the research methods and techniques used, they did not precisely justify the choice of these and not other methods.

Thirdly, it should be emphasized in a more decisive way that the small sample size makes it impossible to generate any conclusions for a wider population. This is a study devoid of statistical significance.

Reviewer 2 Report

My comments on " Dating App Use and Wellbeing: An Application-Based Study Employing Ecological Momentary Assessment and Objective Measures of Use" (ijerph-2092025) are as follows.

The aim of this research was to test the relationship between dating app users’ wellbeing and objective measures of their use during a one-week period. The topic of the manuscript is very interesting. I attempt to put forward only few suggestions to improve the manuscript.

-- First of all, the introduction is weak. Needs a conceptual/ theoretical framework to better interpret current findings.  

--Secondly, due to this sample size is too small, the results of this article may be unstable. So authors need to validate the model's stability.

--Thirdly, in my opinion, the main limitation of the research is that three wellbeing variables are analysed through a single item. I understand that this solution may have been preferred considering the (online) modality and pilot nature of the research, however, Authors should emphasize more this limitation in the dedicated section.

Reviewer 3 Report

The present study explored the relationship between wellbeing measures including self-esteem, mood, and craving, and objective measures of dating app use (i.e., usage time, number of notifications, number of launches). The introduction is well organized and the literature review is updated and related to the subject under investigation. The aim is clear-cut and the hypotheses well-defined. Methods and results are presented in detail. All findings associated with the initial hypotheses are discussed compared to recent literature. Despite some limitations (e.g. the very small sample) mentioned already by the authors, I believe it is a study that deserves to be published. 
